# DarkSAM: Fooling Segment Anything Model to Segment Nothing

**Ziqi Zhou**[1,2,3*], **Yufei Song**[†], **Minghui Li**[‡], **Shengshan Hu**[1,2,4,5†], **Xianlong Wang**[1,2,4,5†],
**Leo Yu Zhang**[§], **Dezhong Yao**[1,2,3*], **Hai Jin**[1,2,3*]
[1] National Engineering Research Center for Big Data Technology and System
[2] Services Computing Technology and System Lab     [3] Cluster and Grid Computing Lab
[4] Hubei Engineering Research Center on Big Data Security
[5] Hubei Key Laboratory of Distributed System Security
∗ School of Computer Science and Technology, Huazhong University of Science and Technology
† School of Cyber Science and Engineering, Huazhong University of Science and Technology
‡ School of Software Engineering, Huazhong University of Science and Technology
§ School of Information and Communication Technology, Griffith University
`{zhouziqi,yufei17,minghuili,hushengshan,wxl99,dyao,hjin}@hust.edu.cn`
`leo.zhang@griffith.edu.au`

## Abstract

*Segment Anything Model* (SAM) has recently gained much attention for its outstanding generalization to unseen data and tasks. Despite its promising prospect, the vulnerabilities of SAM, especially to *universal adversarial perturbation* (UAP) have not been thoroughly investigated yet. In this paper, we propose DarkSAM, the first prompt-free universal attack framework against SAM, including a semantic decoupling-based spatial attack and a texture distortion-based frequency attack. We first divide the output of SAM into foreground and background. Then, we design a shadow target strategy to obtain the semantic blueprint of the image as the attack target. DarkSAM is dedicated to fooling SAM by extracting and destroying crucial object features from images in both spatial and frequency domains. In the spatial domain, we disrupt the semantics of both the foreground and background in the image to confuse SAM. In the frequency domain, we further enhance the attack effectiveness by distorting the high-frequency components (*i.e.*, texture information) of the image. Consequently, with a single UAP, DarkSAM renders SAM incapable of segmenting objects across diverse images with varying prompts. Experimental results on four datasets for SAM and its two variant models demonstrate the powerful attack capability and transferability of DarkSAM. Our codes are available at: https://github.com/CGCL-codes/DarkSAM.

## 1   Introduction

With the advancement of deep learning, large language models, such as GPT [2], LaMDA [33], and PaLM [6], have achieved tremendous success, yet the development of large vision models lags behind. Recently, *Segment Anything Model* (SAM) [19] was proposed as a foundational vision model, demonstrating exceptional generalization capabilities for handling complex segmentation tasks. Unlike traditional segmentation models [24, 42] that output pixel-level labels, SAM introduces a novel prompt-guided image segmentation paradigm by directly producing label-free masks for object segmentation. Benefiting from its powerful zero-shot capability, SAM has been rapidly deployed across various downstream scenarios, such as medical images [34], videos [36], and 3D point clouds [12].

38th Conference on Neural Information Processing Systems (NeurIPS 2024).

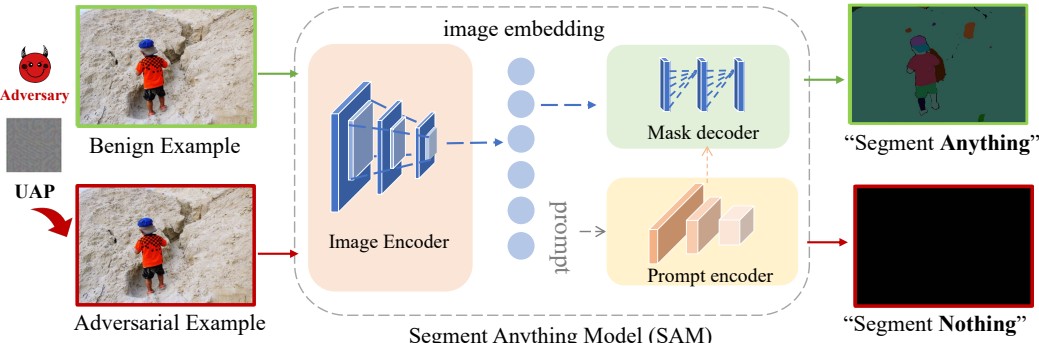

Figure 1: Illustration of fooling SAM using UAP

*Deep neural networks* (DNNs) are shown to be vulnerable to adversarial examples [16, 21, 41, 46], and SAM is no exception. Standard adversarial attacks are designed for classification tasks and cause misclassification by manipulating global image features through image-level perturbations. Existing attacks can be divided into crafting sample-wise adversarial perturbation [25] and *universal adversarial perturbation* (UAP) [27]. The former is tailored for specific inputs, while the latter seeks a single perturbation applicable across a wide range of inputs, thereby intensifying its complexity and difficulty. As a pioneering prompt-guided segmentation model, SAM relies on both *input images* and *prompts* to yield *label-free* masks, rendering existing adversarial attacks [1, 10, 25, 26] focusing only for images and relying on labels ineffective.

Recent efforts [17, 39] started to explore the robustness of SAM against sample-wise adversarial perturbations. Attack-SAM [39] employs classical FGSM [10] and PGD [25] to remove or manipulate the predicted mask for a given image and prompt pair. Meanwhile, another study [17] also investigates the robustness of SAM against various adversarial attacks and corrupted images. However, the more challenging universal adversarial attacks, which more closely resemble real-world scenarios, remain far less thoroughly explored. The introduction of extra and varying prompts in SAM's input, coupled with the lack of label information in its output for attack optimization, renders attacking SAM exceedingly challenging, posing an intriguing problem:

> *Is it feasible to fool the Segment Anything model to segment nothing through a single UAP?*

In this paper, we take a substantial step towards bridging the gap between SAM and UAP. We propose DarkSAM, the first truly prompt-free universal adversarial attack on the prompt-guided image segmentation models (*i.e.*, SAM and its variants), aiming to disable their segmentation ability across diverse input images using a single UAP, irrespective of prompts (see Fig. 1). Unlike classification models that focus on global features, prompt-guided segmentation models concentrate more on local critical objects within images (*e.g.*, objects indicated by prompts). Therefore, our intuition is to destroy crucial object features in the image to mislead SAM into incorrectly segmenting the input images. To this end, DarkSAM is dedicated to decoupling the crucial object features of images from both spatial and frequency domains, utilizing a UAP to disrupt them. 1) In the spatial domain, we begin by dividing SAM's output into foreground (*i.e.*, positive mask values) and background (*i.e.*, negative mask values) via a Boolean mask. We then scramble SAM's decision by destroying the features of the foreground and background of the image, respectively. 2) In the frequency domain, inspired by the factor that SAM is biased towards image texture over shape [38], we employ a frequency filter to decompose images into *high-frequency components* (HFC) and *low-frequency components* (LFC). By increasing the dissimilarity in the HFC of adversarial and benign examples while maintaining consistency in their LHC, we further enhance the effectiveness and transferability of UAP. Experimental results on four segmentation benchmark datasets for SAM and its two variant models, HQ-SAM [18] and PerSAM [40], demonstrate that DarkSAM achieves high attack success rates and transferability.

Our main contributions are summarized as follows:

- We propose DarkSAM, the first truly universal adversarial attack against SAM. We employ a single perturbation to prevent SAM from segmenting objects across a range of images under any form of prompt, which further unveils its vulnerability.

- We design a brand-new prompt-free hybrid spatial-frequency universal attack framework against the prompt-guided image segmentation models to generate a UAP thus making them segment nothing, which consists of a semantic decoupling-based spatial attack and a texture distortion-based frequency attack.
- We conduct extensive experiments on four datasets for SAM and its two variant models. Both the qualitative and quantitative results demonstrate that DarkSAM achieves high attack success rates and transferability.

## 2 Background and Related Works

### 2.1 Prompt-guided Image Segmentation

Segment Anything Model [19] is a cutting-edge advancement in computer vision, garnering widespread attention [3, 5, 20, 22, 34] for its powerful segmentation capabilities. Recent works have been dedicated to exploring various variants of SAM to further enhance performance, such as HQ-SAM [18], PerSAM [40], and MobileSAM [37]. Distinct from traditional semantic segmentation models [4, 24, 42] that predominantly focus on pixel-level label prediction, SAM undertakes the label-free mask prediction by generating object masks for a wide array of subjects using prompts. It consists of three components: an image encoder, a prompt encoder, and a lightweight mask decoder. The image encoder generates image representations in latent space and the prompt encoder utilizes positional embeddings for representing prompts, such as points and boxes. The mask decoder, combining outputs from both image and prompt encoders, predicts effective masks to segment targeted objects.

Given an image $x \in \mathbb{R}^{H \times W \times C}$ and a corresponding prompt $\mathbb{P}$ to SAM, denoted as $f(x, \mathbb{P}) \in \mathbb{R}^{H \times W}$, the model returns a mask $m$ with the predicted segmentation. The prediction process of SAM can be represented as follow:

$$m = f_\theta(x, \mathbb{P}), \tag{1}$$

where $\theta$ represents the parameter of $f(\cdot)$. For an image $x$, each pixel located at coordinates $(i, j)$, referred to as $x_{ij}$, is deemed a part of the masked region when its corresponding mask value $m_{ij}$ exceeds a defined threshold of zero. Recent exploratory studies [39, 17, 35] have revealed vulnerabilities of SAM to adversarial examples and common image corruptions. Different from previous works, our goal is to develop a powerful universal adversarial attack for such prompt-guided image segmentation models.

### 2.2 Universal Adversarial Perturbation

Deep neural networks have been shown vulnerable to adversarial examples [10, 25, 44, 45, 46], where attackers can deceive models by introducing subtle noise to images. *Universal adversarial perturbation* [27] (UAP) was first proposed to fool the victim model by imposing a single adversarial perturbation on a series of images. Existing works can be divided into data-dependent universal adversarial attacks [14, 27, 30] and data-free universal attacks [28, 29, 31], both designed for classification attacks. The former relies on the specific data characteristics of target dataset for UAP generation, while the latter provides a more generalized approach without relying on such data. Meanwhile, some works [15] have also explored UAPs for traditional segmentation models, but they rely on pixel-level labels, which are not applicable to emerging prompt-guided segmentation models. The concurrent works [8, 13] explore UAPs against SAM from the perspectives of direct noise optimization and perturbing the output of the image encoder of SAM, respectively. Different from them, we aim to comprehensively decouple and disrupt crucial object features in images from both spatial and frequency domains, thereby deceiving SAM into failing to segment input images.

## 3 Methodology

### 3.1 Problem Formulation

As a fundamental vision model, SAM typically operates in an online mode, allowing users to set prompts randomly. Therefore, we define the threat model as a quasi-black-box setting, where ad-

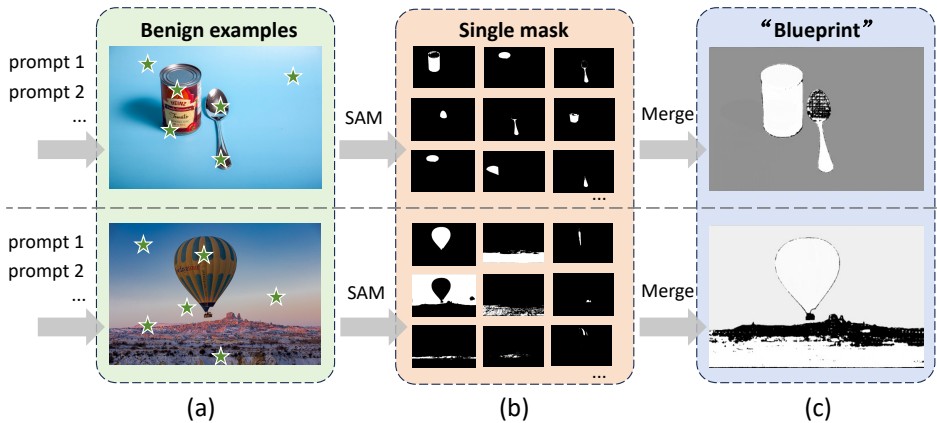

Figure 2: Illustration of the proposed shadow target strategy

versaries have access to the official open-source SAM, but not to the pre-training dataset and the downstream dataset (*i.e.*, those used by users). The adversaries' goal is to craft a UAP $\delta$ using a surrogate dataset $\mathcal{D}_s$ (*i.e.*, unrelated to the pre-training and downstream dataset), thereby compromising the model's performance, *i.e.*, rendering adversarial examples unable to be correctly segmented by SAM. Additionally, the $\delta$ should be sufficiently small, and constrained by $l_p$-norm of $\epsilon$. This problem can be formulated as:

$$\max_{\delta} \mathbb{E}_{x \sim \mathcal{D}_s} \left[ f_\theta \left( x + \delta, \mathbb{P} \right) \neq f_\theta \left( x, \mathbb{P} \right) \right], s.t. \|\delta\|_p \leq \epsilon. \tag{2}$$

## 3.2 Intuition Behind DarkSAM

Unlike the standard deep learning paradigm that inputs a single image and outputs a one-hot label or pixel-level label, SAM requires both images and prompts as inputs, and then outputs label-free masks, indicating the shape information of critical objects. Therefore, a truly universal adversarial attack against SAM should implement a single perturbation to achieve ineffective segmentation for any combination between a series of images and different prompts. However, this task is hindered by the following challenges:

**Challenge I: The dual ambiguity in attack targets arising from varying images and prompts.** Previous UAP works only need to optimize in the target images, hence the introduction of prompts could lead to invalid attacks, as different prompts for a fixed image yield distinct segmentation results. For instance, the image in the top-left corner of Fig. 2 shows a can and a spoon. For the same image, feeding different prompts will result in different masks output by SAM (see Fig. 2(b)). In conclusion, diverse variations in target images and prompts increase the uncertainty of attack targets. For varying images, existing UAP solutions (*e.g.*, UAPGD [9]) can provide references, and the main challenge here is the uncertainty of the attack target brought by unknown prompts. To this end, we propose a *shadow target strategy* by increasing the number of prompts during the attack process to enhance the cross-prompt transferability of UAP. Specifically, for a given input image, we randomly select $k$ prompts (*e.g.*, points or boxes) to create a prompt auxiliary set. By merging their masks output by SAM, we form a *semantic blueprint* of the image, which serves as the target for our attack, as illustrated in Fig. 2(c). This semantic blueprint effectively encompasses the main semantic content of the original image, substantially reducing the ambiguity associated with unknown prompts.

**Challenge II: Suboptimal attack efficacy due to semantic decoupling deficiency.** Since prompt-guided segmentation models output masks that are neither one-hot nor pixel-level labels, traditional attack methods that rely on label deviation for optimization guidance become ineffective. Another approach involves directly modifying the output, such as adjusting the adversarial examples' masks to diverge from their originals, potentially yielding marginal attack success as verified in Sec. 4.4. Nonetheless, the intrinsic sensitivity of segmentation models to pixel-level details significantly constrains the potency of these attacks, underscoring a notable limitation in their applicability.

Given the focus of prompt-guided segmentation models on local, critical object features rather than global image features, we are motivated to comprehensively decouple the key semantic features

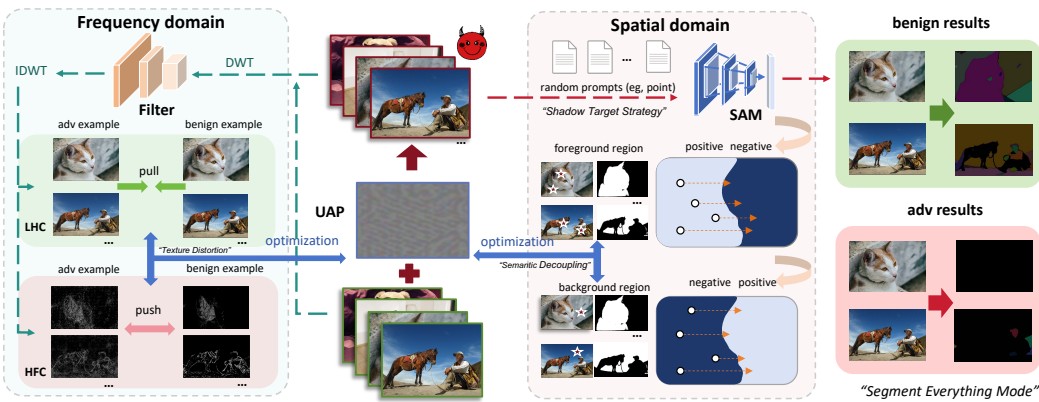

Figure 3: The framework of DarkSAM

of an image from the perspective of both spatial and frequency domains, aiming to fool SAM by manipulating these features. We first define the main object within the image (*i.e.*, the target of segmentation, typically a region rich in texture) as the *foreground*, with the rest being defined as the *background*. As the mask output by SAM indicates the foreground with positive values and the background with negative ones, we use a Boolean mask to separately extract these foreground and background masks. Subsequently, we optimize the UAP, switching adversarial examples' foreground to negative and background to positive, disrupting the image's semantics for a spatial attack. At the same time, inspired by the recent study [38] that SAM is biased towards texture of the image over shape, we investigate the alteration of the high-frequency components (*i.e.*, texture information) of adversarial examples in the frequency domain, while simultaneously constraining the low-frequency components (*i.e.*, shape information), in order to further enhance the effectiveness and transferability of our attack. By separately decoupling and destroying crucial features in both the spatial and frequency domains, we provide valuable optimization directions for UAP generation, thereby facilitating effective attacks on SAM.

### 3.3 DarkSAM: A Complete Illustration

In this section, we present DarkSAM, a novel prompt-free hybrid spatial-frequency universal adversarial attack against the prompt-guided image segmentation models (*i.e.*, SAM and its variants). The pipeline of DarkSAM is depicted in Fig. 3, encompassing a semantic decoupling-based spatial attack and a texture distortion-based frequency attack. We start by randomly generating $k$ different prompts to form an auxiliary prompt set $\mathbb{P}_a$, acquiring the semantic blueprints of the target images as the attack targets. By individually manipulating the semantic content of adversarial examples' foreground and background in the spatial domain, and increasing the distance between the HFC of adversarial and benign examples in the frequency domain, while also constraining the difference in their LFC, we enhance the attack performance and transferability of the UAP. We provide the detailed optimization process of DarkSAM in Algorithm 1. The overall optimization objective $\mathcal{J}_{total}$ of DarkSAM is as follow:

$$\mathcal{J}_{total} = \mathcal{J}_{sa} + \lambda \mathcal{J}_{fa}, \qquad (3)$$

where $\mathcal{J}_{sa}$ and $\mathcal{J}_{fa}$ are the spatial and frequency attack losses, and $\lambda$ controls the importance.

**Semantic decoupling-based spatial attack.** Initially, we utilize two Boolean mask $m_{fg}$ and $\overline{m_{fg}}$ to separately extract the foreground and background mask of the adversarial examples based on the positive and negative values in the mask output by SAM. As for the foreground, our intention is to render it unidentifiable and unsegmentable by SAM. Thus, we optimize its mask towards a negative fake mask $\xi_{neg}$, enabling its fusion with the background to achieve segmentation evasion. The foreground evasion loss $\mathcal{J}_{fe}$ can be described as:

$$\mathcal{J}_{fe} = \mathcal{J}_d(f_\theta(x + \delta, \mathbb{P}_a) \cdot m_{fg}, \xi_{neg}), \qquad (4)$$

where $\xi_{neg}$ is a fake mask that conforms to the shape of the image, containing threshold values of $-\tau$ in regions corresponding to the foreground, and $0$ elsewhere. $\mathcal{J}_d$ serves as the distance metric function, representing the mean squared error loss. For the background, we optimize its mask towards a positive fake mask $\xi_{pos}$ (opposite to $\xi_{neg}$), misleading SAM into interpreting it as a

semantically meaningful object, consequently causing further interference in the assessment of the foreground. The associated loss is

$$\mathcal{J}_{bm} = \mathcal{J}_d(f_\theta(x + \delta, \mathbb{P}_a) \cdot \overline{m_{fg}}, \xi_{pos}). \tag{5}$$

The loss of the semantic decoupling-based spatial attack can be expressed as:

$$\mathcal{J}_{sa} = \mathcal{J}_{fe} + \mathcal{J}_{bm}. \tag{6}$$

**Texture distortion-based frequency attack.** In the frequency domain, the high-frequency components of an image denote the finer details, including noise and textures, while the low-frequency components contain the general outline and overall structural information of the image. We employ the *discrete wavelet transform* (DWT), utilizing a low-pass filter $\mathcal{L}$ and a high-pass filter $\mathcal{H}$ to decompose the image $x$ into different components, constituting a low-frequency component $c_{ll}$, a high-frequency component $c_{hh}$, and two mid-frequency components $c_{lh}$ and $c_{hl}$, via

$$c_{ll} = \mathcal{L}x\mathcal{L}^T, c_{hh} = \mathcal{H}x\mathcal{H}^T, c_{lh}/c_{hl} = \mathcal{L}x\mathcal{H}^T/\mathcal{H}x\mathcal{L}^T. \tag{7}$$

Subsequently, we employ the *inverse discrete wavelet transform* (IDWT) to reconstruct the signal that have been decomposed through DWT into an image. We choose the LFC and HFC while dropping the other components to obtain the reconstructed images $\phi(x)$ and $\psi(x)$ as

$$\phi(x) = \mathcal{L}^T c_{ll} \mathcal{L} = \mathcal{L}^T(\mathcal{L}x\mathcal{L}^T)\mathcal{L}, \tag{8}$$

$$\psi(x) = \mathcal{H}^T c_{hh} \mathcal{H} = \mathcal{H}^T(\mathcal{H}x\mathcal{H}^T)\mathcal{H}. \tag{9}$$

By adding a UAP to the images, we alter their HFC, disrupting the original texture information. Simultaneously, we enforce constraints on the low-frequency disparities between adversarial and benign examples to redirect a larger portion of the perturbation towards the high-frequency domain. As a result, we enhance the attack performance and cross-domain transferability of the UAP by introducing variations in the frequency domain. The loss of texture distortion-based frequency attack can be expressed as:

$$\begin{aligned} \mathcal{J}_{fa} &= \mathcal{J}_{lfc} - \mu\mathcal{J}_{hfc} \\ &= \mathcal{J}_d(\phi(x), \phi(x + \delta)) - \mu\mathcal{J}_d(\psi(x), \psi(x + \delta)), \end{aligned} \tag{10}$$

where $\mu$ is a pre-defined hyperparameter.

---

**Algorithm 1** DarkSAM

---

**Input:** image $x \in \mathcal{D}_s$, SAM $f(x)$ with parameter $\theta$, hyper parameters $k$, $\tau$, $\lambda$, and $\mu$, max-perturbation constraint $\epsilon$
**Output:** A universal adversarial perturbation $\delta$
 1: Initialize random prompt sets $\mathbb{P}_a$ and noise $\delta_o$
 2: Initialize adversarial examples: $x^* \longleftarrow x + \delta_o$
 3: Project $x^*$ to $[0, 1]$ via clipping
 4: Separating different frequency components of $x$ using the discrete wavelet transform:
    $c_{ll}, c_{lh}, c_{hl}, c_{hh} \longleftarrow DWT(x)$
 5: Restore part of the frequency components into an image using the inverse discrete wavelet transform: $\phi(x), \psi(x) \longleftarrow IDWT(c_{ll}), IDWT(c_{hh})$
 6: Calculate $m_{fg}$ by determining the sign of each value in output of $f_\theta(x, \mathbb{P}_a)$
 7: **while** max iterations or not converge **do**
 8:     Calculate spatial loss mentioned in Eq. 6
 9:     Calculate frequency loss mentioned in Eq. 10
10:     Update $\delta$ through backprop
11:     Clip $\delta$ to satisfy imperceptibility constraint $\epsilon$
12:     Project $x^*$ to $[0, 1]$ via clipping
13: **end while**

---

Table 1: The mIoU (%) of DarkSAM under different settings. Values covered by gray denote the clean mIoU, others denote adversarial mIoU. ADE20K, MS-COCO, CITYSCAPES abbreviated as ADE, COCO, CITY, respectively. Bolded values indicate the best results.

| Setting | | SAM [19] | | | | HQ-SAM [18] | | | | PerSAM [40] | | | |
|---|---|---|---|---|---|---|---|---|---|---|---|---|---|
| Prompt | Surrogate | ADE | COCO | CITY | SA-1B | ADE | COCO | CITY | SA-1B | ADE | COCO | CITY | SA-1B |
| POINT | **Clean** | 65.39 | 62.79 | 50.70 | 77.21 | 63.39 | 65.38 | 50.25 | 72.89 | 64.61 | 62.91 | 51.22 | 77.93 |
| | ADE | 0.43 | 3.21 | **0.02** | 5.81 | 0.99 | 6.04 | 6.82 | 7.75 | 0.37 | 3.25 | 7.95 | 1.54 |
| | COCO | 0.42 | 1.16 | 0.76 | 2.46 | **0.69** | **2.23** | 3.19 | 3.46 | **0.01** | **0.05** | 2.56 | 0.07 |
| | CITY | 10.54 | 22.23 | 0.07 | 22.11 | 9.95 | 25.12 | 3.01 | 20.82 | 0.93 | 5.57 | **0.43** | 1.19 |
| | SA-1B | **0.24** | **0.91** | 0.04 | **0.14** | 1.20 | 5.74 | **2.81** | **0.75** | 0.05 | 0.07 | 2.54 | **0.05** |
| | **AVG** | 2.91 | 6.88 | 0.22 | 7.63 | 3.21 | 9.78 | 3.96 | 8.20 | 0.34 | 2.24 | 3.37 | 0.71 |
| BOX | **Clean** | 74.59 | 79.00 | 64.60 | 89.41 | 72.95 | 81.19 | 62.00 | 86.80 | 72.30 | 78.83 | 64.46 | 89.32 |
| | ADE | 4.87 | 10.74 | 1.64 | 14.81 | 3.77 | 12.95 | 19.16 | 20.90 | 2.32 | 8.82 | 19.33 | 14.06 |
| | COCO | **1.51** | **2.97** | 1.96 | 9.22 | **0.27** | **9.38** | 9.74 | 20.97 | **0.41** | **1.38** | 12.04 | 3.20 |
| | CITY | 17.39 | 26.43 | **0.33** | 16.10 | 4.43 | 20.60 | **3.82** | 27.66 | 3.09 | 13.40 | **2.49** | 35.25 |
| | SA-1B | 16.81 | 27.38 | 9.19 | **5.19** | 10.16 | 24.90 | 18.06 | **1.01** | 5.49 | 17.12 | 16.46 | **0.81** |
| | **AVG** | 10.15 | 16.88 | 3.28 | 11.33 | 4.66 | 16.96 | 12.70 | 17.64 | 2.83 | 10.18 | 12.58 | 13.33 |

# 4 Experiments

## 4.1 Experimental Setup

**Datasets and models.** We evaluate our method using four public segmentation datasets: ADE20K [43], MS-COCO [23], CITYSCAPES [7], and SA-1B [19]. For each dataset, we randomly select 100 images for UAP generation and 2,000 images for testing purposes. All images are uniformly resized to $3\times1024\times1024$. For victim models, we use the pre-trained SAM [19], HQ-SAM [18] and PerSAM [40] with the ViT-B backbone.

**Parameter setting.** Following [9, 27, 32], we set the upper bound of UAP to $10/255$. For our experiments, we adjust the hyperparameters $k$, $\tau$, $\lambda$, and $\mu$ to 10, 1, 0.1, and 0.01, respectively, and set the batch size to 1. To evaluate the cross-prompt attack capabilities of DarkSAM, we employ three distinct prompt types: point, box, and segment everything (also abbreviated as "all") mode.

**Evaluation metrics.** To evaluate the effectiveness of DarkSAM, we use the *mean Intersection over Union* (mIoU) metric. To facilitate data presentation, we also use the *attack success rate* (ASR) as a metric to evaluate attack performance. ASR represents the difference between the mIoU values of benign and adversarial examples.

## 4.2 Attack Performance

To comprehensively evaluate DarkSAM's effectiveness, we perform experiments on three prompt-guided image segmentation models including SAM, HQ-SAM, and PerSAM, across four datasets. For each setup, we generate UAPs using point and box prompts, respectively, and then evaluate DarkSAM's attack performance using the corresponding single-point or single-box prompt. We first calculate the clean mIoU of different models across four datasets using point and box as prompts. Specifically, for the SA-1B dataset, we directly extract point and box prompts from the annotations, whereas for the other datasets, we obtain internal points and external boxes as prompts by calculating the object contour coordinates within their annotations.

The experiments in Tab. 1 show that DarkSAM can effectively fool these prompt-guided image segmentation models with an average mIoU reduction of more than $60\%$ across 96 different experimental settings. The results in Tab. 1 also indicate that box prompts not only yield higher segmentation accuracy but also demonstrate greater robustness. For adversaries, the choice of surrogate datasets has a minor impact on crafting UAPs, yet they consistently facilitate excellent attack performance. Notably, DarkSAM demonstrates a distinct advantage when the SA-1B dataset, the training data for SAM, is employed as the surrogate dataset. In addition to the above **quantitative** experimental results, we also present **qualitative** findings. Specifically, we provide the visualization of SAM segmentation results for adversarial examples made by DarkSAM using point and box prompts across four different datasets in Fig. 4. These results include masks of objects in images output by SAM under point, box, and segment everything prompt modes. From Fig. 4, we can see that SAM successfully segments benign images across three types of prompt modes, but it is unable to segment adversarial examples, *i.e.*, the output masks are close to "dark". The qualitative results further corroborate the powerful attack capability of DarkSAM.

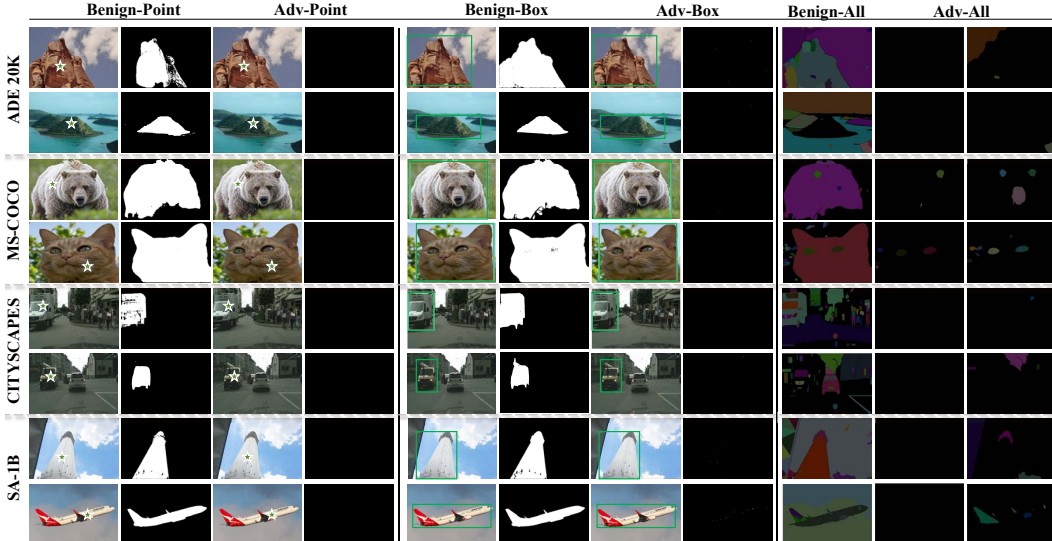

Figure 4: Visualizations of SAM segmentation results for adversarial examples across four datasets. The first four columns and the middle four columns display the segmentation results for point and box prompts, respectively. The last three columns show results under the segment everything mode for benign examples, as well as adversarial examples created using point and box prompts, respectively.

Table 2: The ASR (%) of the cross-prompt transferability study on SAM. "BOX → POINT" indicates that adversarial examples created using box are tested in point mode. Others stand the same meaning.

| Surrogate | BOX → POINT | | | | POINT → BOX | | | |
|---|---|---|---|---|---|---|---|---|
| | ADE | COCO | CITY | SA-1B | ADE | COCO | CITY | SA-1B |
| ADE | 63.01 | 55.61 | 49.72 | 66.67 | **47.00** | **35.52** | 56.15 | 40.80 |
| COCO | **64.95** | **61.69** | 49.98 | **75.09** | 19.95 | 25.27 | 44.60 | 53.01 |
| CITY | 48.31 | 30.48 | **50.30** | 55.74 | 17.36 | 10.94 | 55.43 | 20.69 |
| SA-1B | 52.47 | 36.05 | 47.12 | 66.20 | 31.16 | 17.45 | **58.21** | **62.00** |
| **AVG** | 57.19 | 45.96 | 49.28 | 65.93 | 28.87 | 22.30 | 53.60 | 44.13 |

## 4.3 Transferability Study

We study the attack transferability of DarkSAM across data domain, prompt types, and models, respectively. ❶ **Cross-domain.** The results in Tab. 1 demonstrate DarkSAM's excellent cross-domain transferability, where UAPs generated with the surrogate dataset (ADE20K) achieve a high ASR on datasets from various different domains. We also explore the role of the frequency attack (*i.e.*, $\mathcal{J}_{fa}$, denoted as FA) in enhancing cross-domain transferability. As shown in Fig. 5 (a), frequency attack can effectively improve the attack performance based on the spatial attack (*i.e.*, $\mathcal{J}_{sa}$, denoted as SA). ❷ **Cross-prompt.** We examine the performance of DarkSAM across various types of prompts. As demonstrated in the last three columns of Fig. 4, UAPs created based on both point and box prompts perform well under the segment everything mode. Additionally, we provide results of transferability experiments between point and box prompts in Tab. 2. This includes testing UAPs created with point prompts in the box prompt setting and vice versa. Based on the observed results, it is discernible that UAPs crafted using box prompts generally demonstrate better transferability compared to those using point prompts. This increased efficacy can likely be attributed to the box prompts offering more integral and detailed prompt information. ❸ **Cross-model.** We use UAPs created with points and boxes based on SAM to attack HQ-SAM and PER-SAM. The results in Fig. 5 (b) - (e) showcase DarkSAM's exceptional transferability across different models.

## 4.4 Comparison Study

To comprehensively demonstrate the superiority of our proposed method, we compare DarkSAM with popular UAP schemes, including UAP [27], UAPGD [9], and SSP [32]. We also consider the state-of-the-art adversarial attack against traditional segmentation models, SegPGD [11], and the

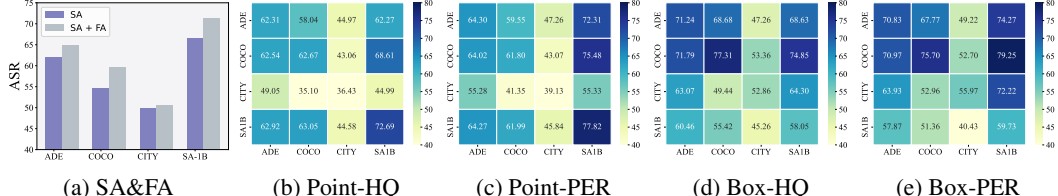

|  | (a) SA&FA | (b) Point-HQ | (c) Point-PER | (d) Box-HQ | (e) Box-PER |

Figure 5: The ASR (%) of transferability study. (a) explores the impact of the frequency attack on boosting the cross-domain transferability of UAPs. (b) - (e) stand the results of cross-model transferability study. "Point-HQ" and "Box-HQ" denote the results of HQ-SAM under point and box prompts, while the suffix "-PER" represents the corresponding results for PerSAM. Each row represents the same UAP.

Table 3: The ASR (%) of comparison study

| Method | POINT → POINT | | | | BOX → BOX | | | |
|---|---|---|---|---|---|---|---|---|
|  | ADE | COCO | CITY | SA-1B | ADE | COCO | CITY | SA-1B |
| UAP [27] | 1.62 | 0.47 | 8.13 | 5.28 | 0.28 | * | 1.29 | 1.76 |
| UAPGD [9] | 4.85 | 1.52 | 11.52 | 10.04 | 0.97 | 0.45 | 2.22 | 3.11 |
| SSP [32] | 0.67 | 0.09 | 5.90 | 4.08 | * | * | 0.91 | 1.20 |
| SegPGD [11] | 4.24 | 1.44 | 11.48 | 8.92 | 0.89 | 0.51 | 2.10 | 3.46 |
| Attack-SAM [39] | 2.91 | 1.36 | 13.20 | 9.54 | 0.51 | 0.36 | 1.90 | 3.12 |
| Ours | **64.96** | **61.63** | **50.63** | **77.07** | **69.72** | **76.03** | **64.27** | **84.22** |

latest sample-wise attack against SAM, Attack-SAM [39]. For a fair comparison, we adapt them to a UAP optimization strategy and keep other settings consistent with DarkSAM. We select SAM as the victim model and assess the effectiveness of these UAP methods across four datasets, using the same dataset for both generating and testing the UAPs. The results in Tab. 3 indicate that Dark-SAM outperforms all methods with a considerable margin. The negative experimental values ("*") indicate that the attack does not work at all. This phenomenon may stem from counterproductive perturbations that inadvertently cause the input samples to resemble the training set used by SAM, paradoxically enhancing accuracy and resulting in negative ASR values. We also provide visualizations of the segmentation results of the adversarial examples made by these methods using box prompts in Fig. 6, obtained in point, box, and segment-everything modes, respectively. The results further demonstrate the superiority of DarkSAM.

## 4.5 Ablation Study

In this section, we explore the effect of different modules, prompt number, attack strengths, training data size, and threshold values on DarkSAM. We conduct experiments using point prompts on SAM across the ADE20K dataset.

**The effect of different modules.** We investigate the effect of various modules on the attack performance of DarkSAM. For clarity and convenience, we use A, B, C, and D to denote $\mathcal{J}_{fe}$, $\mathcal{J}_{bm}$, $\mathcal{J}_{hfc}$, and $\mathcal{J}_{lfc}$, respectively. The results in Fig. 7 (a) show that no variants can compete with the complete method, implying the indispensability of each component for DarkSAM.

**The effect of prompt number.** We study the effect of the prompt number in proposed shadow target strategy on attack performance of DarkSAM. We conduct experiments with varying numbers of point prompts, ranging from 1 to 100. The results in Fig. 7 (b) show a gradual increase in attack performance from 1 to 10 (default setting), followed by a downward trend. This could be attributed to an excess of random points leading to masks with redundant information, thereby impacting the attack efficacy.

**The effect of perturbation budget.** As shown in Fig. 7 (c), we evaluate DarkSAM's attack performance with $\epsilon$ from $4/255$ to $32/255$. With the increase in $\epsilon$, there is a corresponding enhancement in attack performance. Notably, our attack still maintains high efficacy at the $6/255$ setting, with an average ASR exceeding $45\%$.

**The effect of number of training samples.** We explore the effect of varying the number of training images used to create UAP on DarkSAM. Utilizing a range from 10 to 1000 images to craft UAPs, the results in Fig. 7 (d) reveal that employing merely 100 images can achieve excellent attack performance, demonstrating a strong applicability advantage.

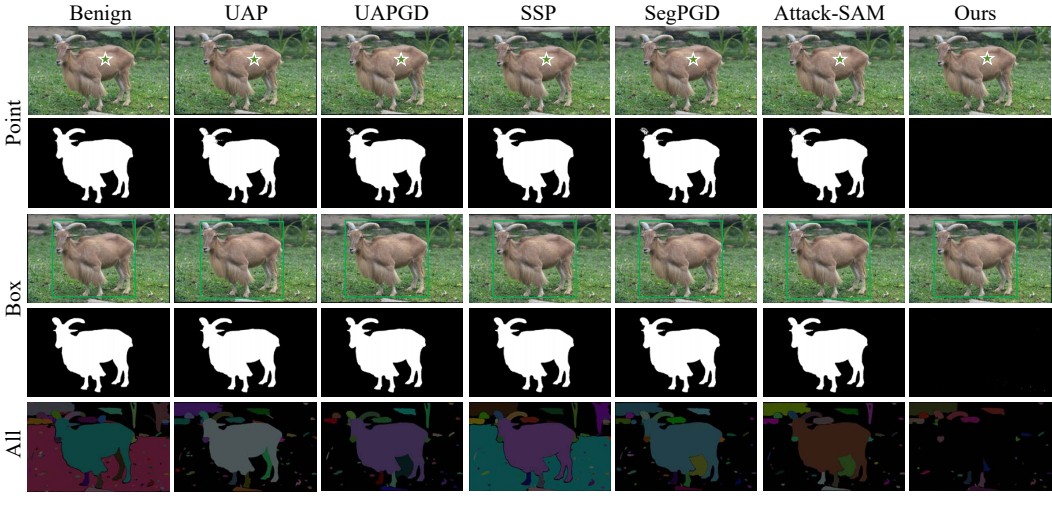

Figure 6: Visualizations of the comparison study

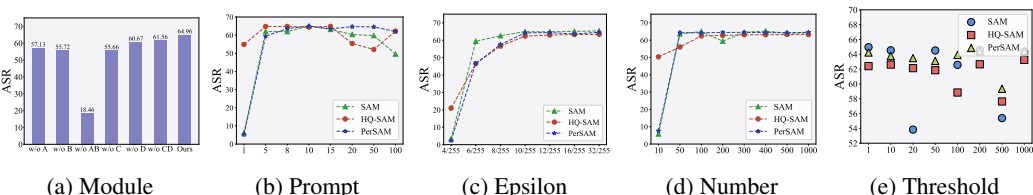

| (a) Module | (b) Prompt | (c) Epsilon | (d) Number | (e) Threshold |

Figure 7: The results (%) of ablation study. (a) - (e) investigate the effect of different modules, prompt number, attack strengths, number of training samples, and threshold values in fake mask on DarkSAM, respectively.

**The effect of threshold values.** We examine the effect of varying threshold values $\tau$ in the fake mask $\xi$ on DarkSAM. As illustrated in Fig. 7 (e), we test a range of values from 1 to 1000. The results indicate that these different values have a minimal overall effect on DarkSAM's performance.

## 5 Conclusions, Limitations, and Broader Impact

In this paper, we propose DarkSAM, the first truly universal adversarial attack against SAM. With a single perturbation, DarkSAM renders SAM incapable of segmenting objects across diverse images with varying prompts, thereby exposing its vulnerability. To tackle the challenge of dual ambiguity in attack targets, we present a shadow target strategy to obtain semantic blueprint as a attack target. We then design a novel prompt-free hybrid spatial-frequency universal attack framework, which consists of a semantic decoupling-based spatial attack and a texture distortion-based frequency attack. By disrupting the crucial object features in both the spatial and frequency domains of the images, it successfully addresses the challenge of suboptimal attack efficacy, thus deceiving SAM. Our extensive experiments on SAM, HQ-SAM, and PerSAM across four datasets, both qualitatively and quantitatively, demonstrate DarkSAM's powerful attack ability and strong attack transferability.

In terms of limitations, DarkSAM may not be suitable for traditional segmentation models because its output is not a label-free mask. This characteristic might limit its applicability in scenarios where labeled masks are essential for accurate segmentation. The adversarial examples produced by DarkSAM could potentially mislead SAM-based segmentation platforms, posing significant security risks, particularly in sensitive domains like medical image analysis.

## Acknowledgements

Shengshan Hu's work is supported by the National Natural Science Foundation of China (Grant Nos. U20A20177, 62372196). Dezhong Yao's work is supported by the National Natural Science Foundation of China under Grant No. 62072204. Shengshan Hu and Dezhong Yao are co-corresponding authors.

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

# Appendix

## A Datasets

- **ADE20K:** ADE20K [43] is a dataset for scene parsing that includes images from a variety of environments. It contains more than 20,000 images, classified into 150 categories, covering both natural landscapes and indoor settings. Each image in ADE20K is pixel-wise annotated, making it suitable for scene parsing and semantic segmentation tasks.
- **MS-COCO:** MS-COCO [23] is a large-scale dataset for image recognition, segmentation, and image captioning. It contains more than 200,000 labeled images, 150,000 validation images, and over 80,000 test images. The dataset includes 80 different object categories and over 250,000 object instances. MS-COCO is known for its detailed annotations for each image, including object segmentation, object detection, and image captioning.
- **CITYSCAPES:** CITYSCAPES [7] is a dataset for urban street scenes, primarily used for training and testing vision systems for autonomous driving. It includes street scenes from 50 different cities, with approximately 5,000 finely annotated images. These images include various urban scenarios and a range of traffic participants.
- **SA-1B:** SA-1B [19] contains 11 million diverse, high-resolution, privacy-protected images and 1.1 billion high-quality segmentation masks. These masks were automatically generated by SAM. The dataset aims to facilitate computer vision research and is characterized by an average of 100 masks per image.

## B Platform

Experiments are conducted on a server running a 64-bit Ubuntu 20.04.1 system with an Intel(R) Xeon(R) Silver 4210R CPU @ 2.40GHz processor, 125GB memory, and two Nvidia GeForce RTX 3090 GPUs, each with 24GB memory. The experiments are performed using the Python language and PyTorch library version 2.1.0.

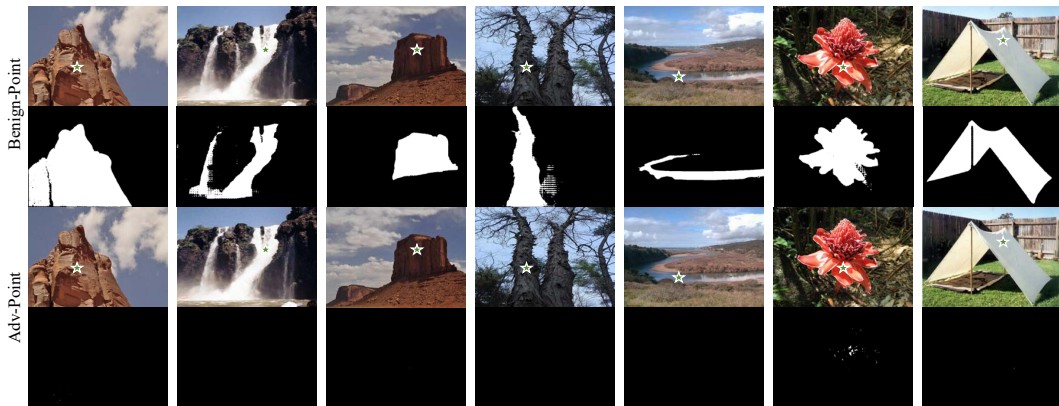

Figure A1: Qualitative results of the DarkSAM using **point** prompts on *SAM-L* under **point** prompts

## C Supplementary Attack Performance

### C.1 Evaluation on MobileSAM

We evaluate the attack performance of DarkSAM against another SAM's variant model, Mobile-SAM [37], on four datasets. All experimental settings are kept consistent with Sec.4.2. The results in Tab. A1 demonstrate the effectiveness of DarkSAM against MobileSAM, further proving its strong attack capability. Notably, in line with the conclusions in Sec.4.2, the choice of surrogate datasets has a certain impact on the attack performance. SA-1B serves as a notably superior surrogate dataset, while CITYSCAPES exhibits comparatively lower performance in certain scenarios. This discrepancy may be attributed to CITYSCAPES' limited scope, which solely encompasses the urban street scene, consequently restricting the transferability of the generated UAPs.

## C.2 Evaluation on SAM with ViT-L backbone

We present both quantitative and qualitative results of DarkSAM on SAM with a ViT-L backbone, denoted as SAM-L. The quantitative findings in Table A2 illustrate the effectiveness of DarkSAM in deceiving SAM-L. Notably, these results indicate that SAM-L exhibits greater robustness compared to SAM-B (SAM with a ViT-B backbone) due to its more intricate network architecture. Additionally, we offer visualization results of DarkSAM's attacks on SAM-L under point, box, and segment everything modes. Figs. A1 and A2 demonstrate that adversarial examples generated based on point prompts effectively mislead SAM-L. Similarly, adversarial examples crafted using box prompts also prove to be successful in deceiving SAM-L, as depicted in Figs. A3 and A4.

Table A1: The mIoU (%) of DarkSAM on MobileSAM. Values covered by  gray  denote the clean mIoU, others denote adversarial mIoU. ADE20K, MS-COCO, CITYSCAPES abbreviated as ADE, COCO, CITY, respectively. Bolded values indicate the best results.

| Model | Surrogate | POINT→POINT | | | | BOX→BOX | | | |
|---|---|---|---|---|---|---|---|---|---|
| | | ADE | COCO | CITY | SA-1B | ADE | COCO | CITY | SA-1B |
| MobileSAM [37] | Clean | 63.77 | 63.08 | 51.13 | 76.82 | 72.69 | 79.15 | 64.3 | 89.14 |
| | ADE | 0.82 | 2.48 | 0.47 | 3.99 | **0.98** | **4.99** | 1.34 | 6.02 |
| | COCO | 0.10 | 0.41 | 0.03 | 0.74 | 1.60 | 6.38 | 1.33 | 8.12 |
| | CITY | 16.73 | 31.85 | 0.10 | 41.47 | 26.81 | 49.38 | **0.81** | 48.74 |
| | SA-1B | **0.06** | **0.34** | **2.3e-6** | **1.8e-5** | 4.67 | 14.89 | 0.84 | **3.03** |
| | AVG | 4.43 | 8.77 | 0.15 | 11.55 | 8.52 | 18.91 | 1.08 | 16.48 |

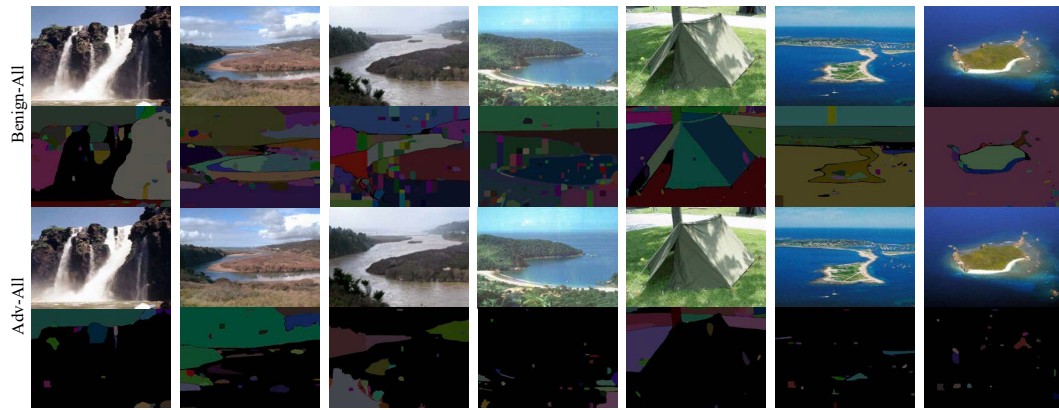

Figure A2: Qualitative results of the DarkSAM using **point** prompts on *SAM-L* under the **segment everything** mode

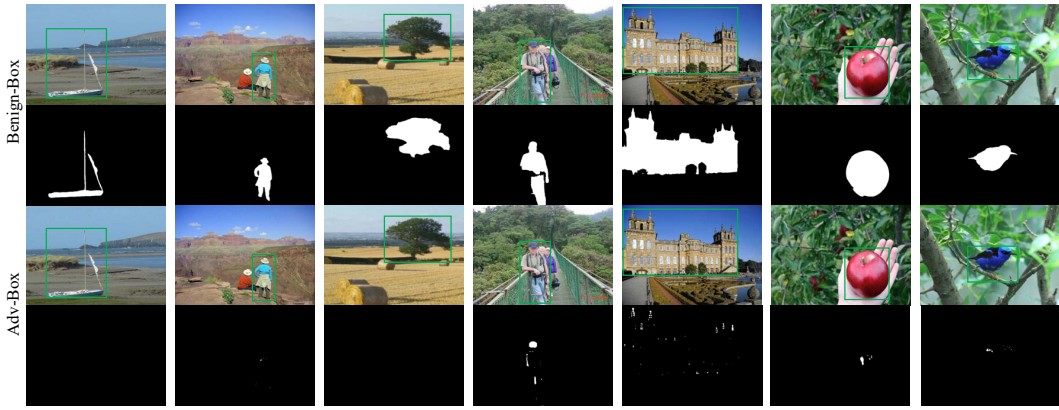

Figure A3: Qualitative results of the DarkSAM using **box** prompts on *SAM-L* under **box** prompts

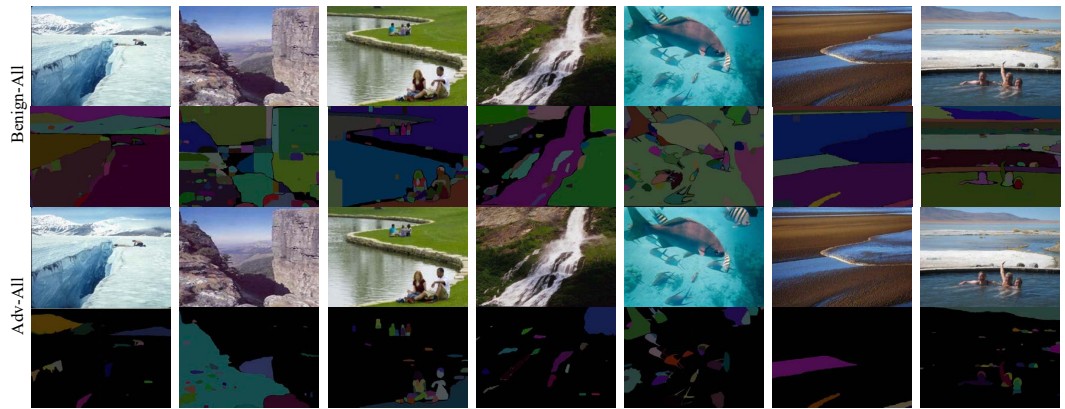

Figure A4: Qualitative results of the DarkSAM using **box** prompts on *SAM-L* under the **segment everything** mode

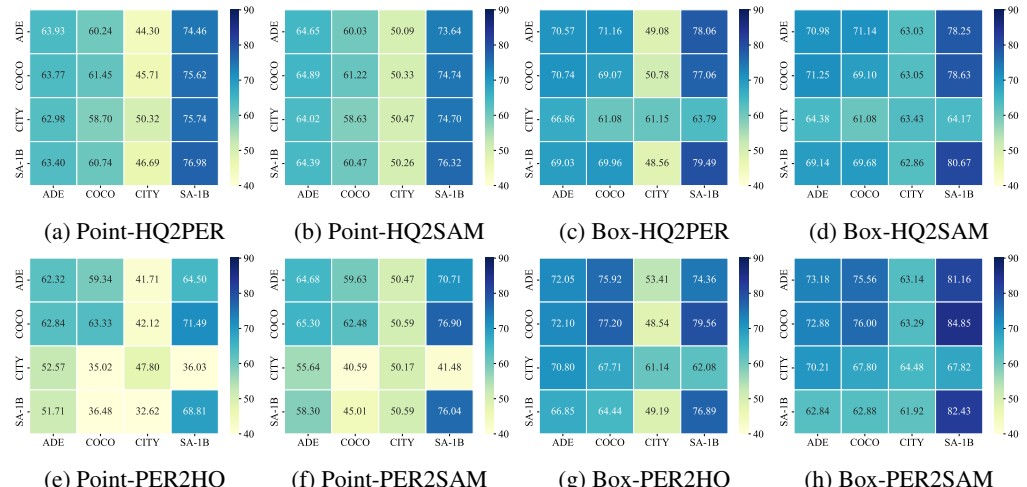

(a) Point-HQ2PER (b) Point-HQ2SAM (c) Box-HQ2PER (d) Box-HQ2SAM

(e) Point-PER2HQ (f) Point-PER2SAM (g) Box-PER2HQ (h) Box-PER2SAM

Figure A5: The ASR (%) of supplementary transferability study. (a) - (d) show the cross-model transferability study results for UAPs created on HQ-SAM. (e) - (h) show the cross-model transferability study results for UAPs created on PerSAM. Each row represents the same UAP.

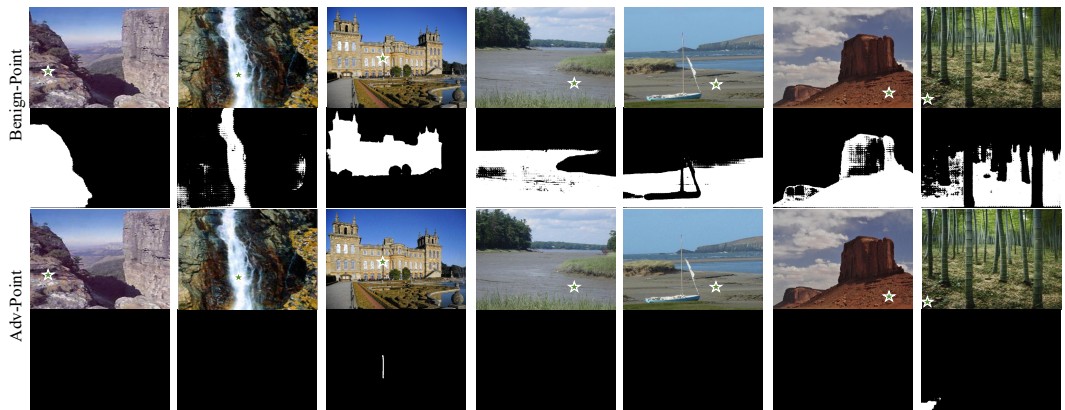

Figure A6: Visualizations of the cross model backbone transferability study. These adversarial examples are all crafted based on *SAM-B* and tested on *SAM-L* under point prompts. (*SAM-B → SAM-L*)

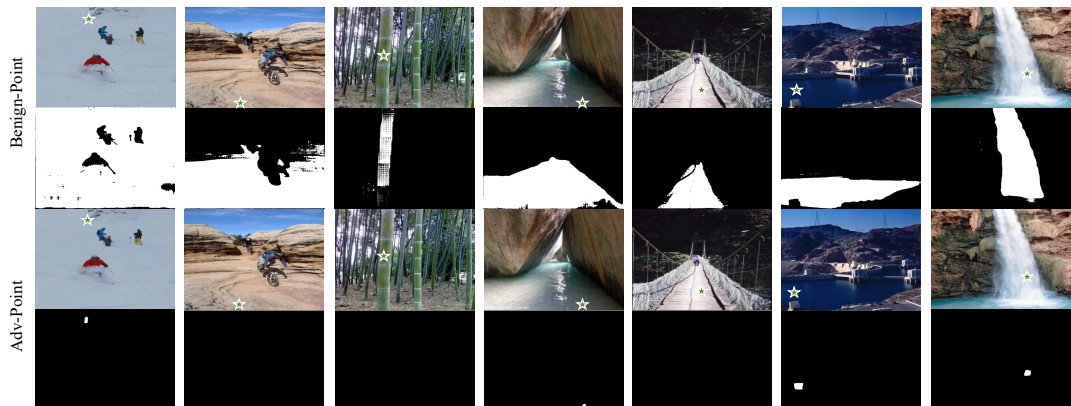

Figure A7: Visualizations of the cross model backbone transferability study. These adversarial examples are all crafted based on *SAM-L* and tested on *SAM-B* under point prompts. (*SAM-L → SAM-B*)

Table A2: The ASR (%) of DarkSAM on *SAM-L*

| Prompt | Surrogate | ADE | COCO | CITY | SA-1B |
|--------|-----------|-------|-------|-------|-------|
| Point | ADE | 43.51 | 49.03 | 23.86 | 43.77 |
| | SA-1B | 34.87 | 38.84 | 42.59 | 48.03 |
| Box | ADE | 49.79 | 48.69 | 47.42 | 43.63 |
| | SA-1B | 62.56 | 66.68 | 51.91 | 52.29 |

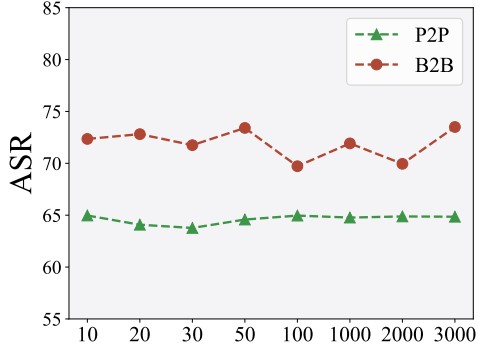

Figure A8: The results (%) of ablation study about random seeds

# D    Supplementary Transferability Study

In this section, we delve deeper into the cross-model transferability of DarkSAM, considering both different model types and diverse model backbones. We maintain uniformity with the experimental settings outlined in Sec. 4.3.

**1) Cross model type transferability.** We explore the transferability of UAPs crafted by DarkSAM on PerSAM and HQ-SAM when attacking other types of models. These models all share the ViT-B backbone. The "Point-" and "Box-"prefixes indicate that the UAPs are crafted and tested under point and box prompts, respectively. The suffixes "HQ2PER" and "HQ2SAM" denote the UAPs crafted on HQ-SAM against PerSAM and SAM, respectively. Similar notations carry the same implications. The results in Fig. A5 further demonstrate the robust cross-model type transferability of DarkSAM.

**2) Cross model backbone transferability.** We investigate the cross-model backbone transferability of DarkSAM. Specifically, we craft UAPs based on the ADE20K dataset on SAM-L and SAM-B, respectively, and test the transferability of these attacks between the two models. From Figs. A6 and A7, we can see that adversarial examples crafted on SAM-B effectively mislead SAM-L, and

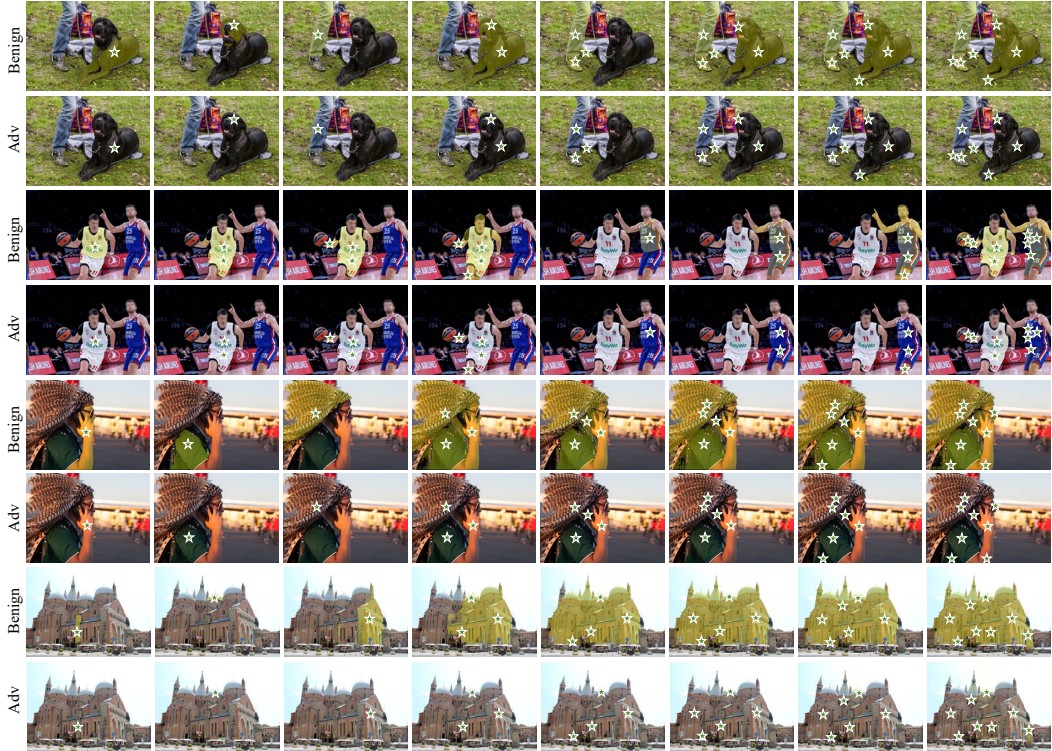

Figure A9: Visualization of the SAM segmentation results of adversarial examples under the multipoint evaluation mode

conversely, those crafted on SAM-L deceive SAM-B. These results demonstrate the cross-model backbone transferability of DarkSAM.

# E Supplementary Ablation Study

**1) The effect of random seeds.** Considering the relationship between random seeds and the selection of images in training and testing, we investigate the effect of random seeds on DarkSAM. All our experiments default to a random seed setting of 100. As illustrated in Fig. A8, we select eight different random seeds and conduct experiments to attack SAM on the ADE20K dataset with these seeds. "P2P" and "B2B" respectively denote the creation and testing of UAPs using point and box prompts. The results in Fig. A8 indicate that DarkSAM consistently exhibits stable and superior attack performance across various random seed settings.

**2) The mixed use of point and box prompts in the *shadow target strategy*.** We delve into the effects of employing a hybrid approach of points and boxes as prompts in the shadow target strategy for DarkSAM. In this method, we craft UAPs using a balanced mix of five random points and five boxes. The outcomes, as detailed in Tab. A3, reveal that UAPs constructed with this mixed approach maintain robust attack performance. This finding accentuates the adaptability and effectiveness of our proposed shadow target strategy.

**3) Multipoint evaluation.** We explore the effects of using multiple point prompts during the inference phase of SAM on the efficacy of DarkSAM's attacks. Compared to single-point prompts, multipoint prompts offer augmented object positional data, potentially enhancing segmentation accuracy. To evaluate this, we generate UAPs on the SA-1B dataset using point prompts and subsequently test these under multipoint prompts mode. The results, as illustrated in Fig. A9, indicate that for benign examples, the use of multipoint prompts indeed results in a more accurate segmentation of the target object relative to single-point prompts. Conversely, in the case of adversarial examples, the addition of multiple prompts does not aid SAM in achieving successful segmentation, thereby underscoring the powerful effectiveness of DarkSAM in compromising segmentation accuracy.

Table A3: The ASR (%) of DarkSAM using **five points** and **five boxes**

| Model | POINT & BOX → POINT | | | | POINT & BOX→ BOX | | | |
|-------|------|------|------|------|------|------|------|------|
| | ADE | COCO | CITY | SA-1B | ADE | COCO | CITY | SA-1B |
| SAM [19] | 64.49 | 61.31 | 50.74 | 76.42 | 73.05 | 71.87 | 63.59 | 86.45 |
| HQ-SAM [18] | 62.49 | 64.43 | 49.09 | 72.26 | 70.65 | 70.08 | 57.49 | 84.28 |
| PerSAM [40] | 63.53 | 62.41 | 50.01 | 76.86 | 70.13 | 76.54 | 62.03 | 85.63 |
| MobileSAM [37] | 63.18 | 62.47 | 50.88 | 76.06 | 70.90 | 75.61 | 63.38 | 85.41 |

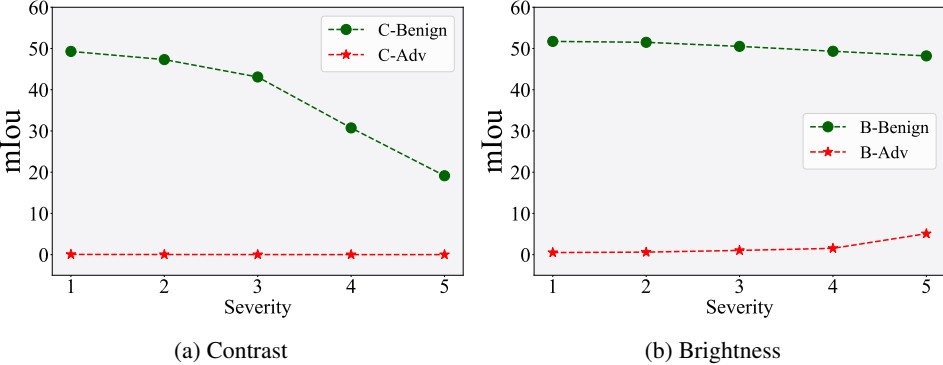

(a) Contrast           (b) Brightness

Figure A10: The results (%) of crruption study (a) investigate the result of Contrast, (b)investigate the result of Brightness

## F Defense

SAM is renowned for its powerful zero-shot capabilities, thus we believe an appropriate defensive measure is to refrain from making additional structural and parametric modifications to the pre-trained SAM to avoid compromising its original knowledge. Therefore, we consider employing input preprocessing methods to counter adversarial examples. We select two famous image corruption methods from the *Imagecorruptions* repository, contrast (C) and brightness (B), to test adversarial examples. Results in Fig. A10 demonstrate that DarkSAM effectively withstands such preprocessing-based defenses.

