# OpenReview forum: "DarkSAM: Fooling Segment Anything Model to Segment Nothing"
_NeurIPS.cc/2024/Conference — NeurIPS 2024 poster_

### Official Review · Reviewer_pM4H · 2024-07-08

**Soundness:** 3
**Presentation:** 3
**Contribution:** 3
**Rating:** 5
**Confidence:** 3

**Summary:**

This paper introduces DarkSAM, a prompt-free universal attack framework against the Segment Anything Model (SAM) in a quasi-black-box setting. The framework consists of a semantic decoupling-based spatial attack and a texture distortion-based frequency attack. While SAM uses geometric prompt inputs to guide segmentation of critical objects within images, DarkSAM disrupts these processes by decoupling the object features of images in both spatial and frequency domains using a universal adversarial perturbation (UAP). In the spatial domain, it scrambles SAM’s decisions by destroying the features of the foreground and background of the image separately. In the frequency domain, it decomposes images into high-frequency components (HFC) and low-frequency components (LFC), increasing the dissimilarity in the HFC of adversarial and benign examples while maintaining consistency in their LFC. Experiments are conducted on four public segmentation datasets (ADE20K, MS-COCO, CITYSCAPES, and SA-1B), with 100 images used for UAP generation and 2,000 images for testing for each dataset. Victim models include the pre-trained SAM, HQ-SAM, and PerSAM with the ViT-B backbone.

**Strengths:**

1. The paper is well-written and easy to follow.
2. The specific design of the shadow target strategy is tailored to SAM for prompt-based segmentation, which is unique compared to adversary attacks against traditional segmentation pipelines.
3. The attacking results are impressive. All three models show very low segmentation performance across multiple datasets.

**Weaknesses:**

SAM is a milestone work, and a series of follow-up studies have been proposed recently. However, this paper does not provide an up-to-date review in Section 2.1 and lacks comparison in Section 4, weakening its significance. For example:
- SAM-based adversary attack:
[1] Practical Region-level Attack against Segment Anything Models, CVPR 2024.
- Other SAM-based models:
[2] From SAM to CAMs: Exploring Segment Anything Model for Weakly Supervised Semantic Segmentation, CVPR 2024.
[3] RobustSAM: Segment Anything Robustly on Degraded Images, CVPR 2024.
[4] Matching Anything by Segmenting Anything, CVPR 2024.
[5] FocSAM: Delving Deeply into Focused Objects in Segmenting Anything, CVPR 2024.
[6] ASAM: Boosting Segment Anything Model with Adversarial Tuning, CVPR 2024.
[7] BA-SAM: Scalable Bias-Mode Attention Mask for Segment Anything Model, CVPR 2024.
[8] Open-Vocabulary SAM: Segment and Recognize Twenty-thousand Classes Interactively, ECCV 2024.
[9] CAT-SAM: Conditional Tuning Network for Few-Shot Adaptation of Segmentation Anything Model, ECCV 2024.
[10] Semantic-SAM: Segment and Recognize Anything at Any Granularity, ECCV 2024.

**Questions:**

1. How does DarkSAM perform with variants of the SAM models?
2. Please include a comparison with [1] "Practical Region-level Attack against Segment Anything Models" (CVPR 2024).

**Limitations:**

The paper presents a limitation that DarkSAM is tailored to SAM and cannot operate on traditional segmentation models.

---

> ### Author Rebuttal · Authors · 2024-08-05
>
> # Responses to Reviewer pM4H #
>
> ------
>
> **Q1**: SAM is a milestone work, and a series of follow-up studies have been proposed recently. However, this paper does not provide an up-to-date review in Section 2.1 and lacks comparison in Section 4, weakening its significance.
>
> **A1**: Thanks for your professional suggestions! We acknowledge that the ten papers provided by the reviewer pM4H are all latest papers from CVPR 2024 (conference dates: June 17-21, 2024) or ECCV 2024 (conference dates: September 29-October 4, 2024), and are therefore contemporaneous with our work. Since **these papers were not officially published** before the NeurIPS 2024 manuscript submission deadline (May 23, 2024), we did not include them in our submission. We are happy to include a discussion and comparison with these recent works in the revised version.
>
> ------
>
> **Q2**: How does DarkSAM perform with variants of the SAM models?
>
> **A2**: We have already evaluated the attack effectiveness of DarkSAM on four different SAM models across four datasets, including SAM [1], HQ-SAM [2], PerSAM [3], and MobileSAM [4] (see Tab. A1 in the Appendix). This is sufficient to demonstrate the effectiveness of our proposed method. Due to time constraints, we select the representative ASAM [5] provided by reviewer pM4H for further testing on four datasets, with results provided in Table R2. The experimental setup is consistent with that described in Sec. 4.2 of the manuscript. These experimental results further demonstrate the high effectiveness of our method against SAM and its variants.
>
> Table R2: The mIoU (%) of DarkSAM on ASAM
>
> | Prompt | Surrogate | ADE   | COCO  | CITY  | SA-1B |
> | ------ | --------- | ----- | ----- | ----- | ----- |
> | Point  | Clean     | 63.57 | 59.73 | 49.07 | 75.17 |
> |        | ADE       | 1.16  | 2.92  | 0.69  | 6.01  |
> |        | COCO      | 1.20  | 2.04  | 1.60  | 4.20  |
> |        | CITY      | 8.97  | 12.07 | 0.17  | 10.90 |
> |        | SA-1B     | 3.17  | 5.43  | 1.21  | 3.04  |
> | Box    | Clean     | 76.28 | 81.26 | 63.91 | 89.13 |
> |        | ADE       | 1.94  | 3.05  | 1.67  | 12.16 |
> |        | COCO      | 1.17  | 2.00  | 0.90  | 3.90  |
> |        | CITY      | 14.41 | 20.36 | 0.39  | 22.24 |
> |        | SA-1B     | 3.02  | 5.33  | 0.76  | 2.79  |
>
> ------
>
> **Q3**: Please include a comparison with "Practical Region-level Attack against Segment Anything Models" (CVPR 2024)
>
> **A3**:   As mentioned in A1, [6] is a contemporaneous study that also explores adversarial attacks against SAM. However, it was not officially published before we submitted our manuscript to NeurIPS 2024. **The study differs from our work in two significant aspects: the attack objective and the attack method.**
>
> 1) **Local vs. Global**: In terms of the attack objective, [6] proposes a region-level attack aimed at concealing objects within a specific attacker-designated region, preventing SAM from segmenting them. In contrast, DarkSAM seeks to completely disable SAM, rendering it unable to segment any object in the input image, regardless of the type of prompt used.
> 2) **Sample-wise vs. Universal**: In terms of the attack method, [6] requires generating sample-specific noise for each image to deceive SAM. In contrast, our approach only requires a single universal adversarial perturbation (UAP) to fool SAM across a range of images, which is a more challenging task.
>
> Due to these significant differences, a direct comparison between [6] and our proposed method is not feasible. To facilitate a fair comparison, one potential approach would be to adapt the method from [6] into a universal adversarial attack format. Unfortunately, [6] does not provide official code, making it difficult to implement and compare their method during the rebuttal period. Although we reached out to the authors immediately upon receiving the review comments, we have not yet received a response. We are willing to include a discussion and comparison with [6] in the revised version.
>
> Additionally, compared to [6], we assess the effectiveness of DarkSAM across a range of experimental conditions, including box prompts (see Tabs. 1-3, A1-A3; Figs. 5, A3, A9, and A10), the segment-everything mode (see Figs. 5, A2, A4, A9, and A10), and multi-point prompts (see Fig. A8). Our extensive and comprehensive experimental results are sufficient to demonstrate the effectiveness and superiority of our method.
>
> **Reference**
>
> [1] Segment Anything,  ICCV 2023.
>
> [2] Segment Anything in High Quality, NeurIPS 2023.
>
> [3] Personalize Segment Anything Model with One Shot, ICLR 2024.
>
> [4] Faster Segment Anything: Towards Lightweight SAM for Mobile Applications, Arxiv 2023.
>
> [5] ASAM: Boosting Segment Anything Model with Adversarial Tuning, CVPR 2024.
>
> [6] Practical Region-level Attack against Segment Anything Models, CVPR 2024.

---

### Official Review · Reviewer_dgw7 · 2024-07-09

**Soundness:** 3
**Presentation:** 3
**Contribution:** 3
**Rating:** 8
**Confidence:** 4

**Summary:**

This work investigates adversarial attacks against Segment Anything Models (SAMs) and presents DarkSAM, the first universal adversarial attack designed for these models. DarkSAM leverages a single perturbation to effectively undermine SAM’s object segmentation capabilities across a variety of images and prompts. The authors conduct a comprehensive evaluation of DarkSAM across four datasets and three SAM variants (SAM, HQ-SAM, and PerSAM), covering attack performance, transferability, comparative analysis, and ablation studies.

**Strengths:**

1.	The paper introduces a unique perspective on adversarial attacks for prompt-guided segmentation models, which is a relatively unexplored area in the literature. The proposed DarkSAM method is innovative in its approach to decoupling object features for attack optimization.
2.	The research question is well-defined, and the authors thoroughly compare DarkSAM with a multitude of established baselines.
3.	The paper presents both qualitative and quantitative results, effectively demonstrating the impact of DarkSAM. These results provide a thorough assessment of its performance across various conditions

**Weaknesses:**

1.	It is recommended that the authors further supplement the experimental section with relevant analyses, such as explaining why the spatial domain attack is more critical than the frequency domain attack within the proposed framework.
2.	In this paper, the usage of mIoU and ASR appears to be analogous, with both metrics conveying the same information. Could the authors provide insight into the justification for employing both metrics concurrently?
3.	In Figure 7, the visualization of segmentation masks is notably dark, impeding discernibility for the reader. The authors should consider increasing the brightness of these images or employing more vivid colors.

**Questions:**

1.	I agree with the authors' prompt-free approach to the attack on SAM. I am curious about the types of prompts that may be more advantageous in crafting effective UAPs during the attack generation process. Additionally, could the authors provide an explanation regarding the selection of prompts?

**Limitations:**

1.	The paper relies on heuristic research and lacks a corresponding theoretical framework for analysis.

---

> ### Author Rebuttal · Authors · 2024-08-05
>
> # Responses to Reviewer dgw7 #
>
> ------
>
> **Q1**: It is recommended that the authors further supplement the experimental section with relevant analyses, such as explaining why the spatial domain attack is more critical than the frequency domain attack within the proposed framework.
>
> **A1**:  Thanks for the constructive suggestion! As observed in Fig. 6(a) of the manuscript, we can see that the spatial domain attack is more critical than the frequency domain attack within the proposed framework. This may be attributed to SAM’s reliance on pixel information in the spatial domain rather than frequency information in the frequency domain during object segmentation. We will include a detailed description of this analysis in the ablation study section of the revised version.
>
> ------
>
> **Q2**: In this paper, the usage of mIoU and ASR appears to be analogous, with both metrics conveying the same information. Could the authors provide insight into the justification for employing both metrics concurrently?
>
> **A2**: We use both mIoU and ASR to evaluate attack performance In this paper. Lower mIoU values and higher ASR indicate stronger attack effectiveness. On one hand, we use mIoU to *quantitatively* and intuitively demonstrate the model's robustness against the proposed method. On the other hand, we employ the *visualization-friendly* ASR to further and comprehensively showcase the superior performance of the proposed approach. These two metrics Together offer a well-rounded assessment of the attack's impact.
>
> ------
>
> **Q3**:  In Figure 7, the visualization of segmentation masks is notably dark, impeding discernibility for the reader. The authors should consider increasing the brightness of these images or employing more vivid colors.
>
> **A3**: Thanks for your valuable feedback! We will enhance the visualization of the segmentation masks to make them more visually appealing in the revised version.
>
> ------
>
> **Q4**: I agree with the authors' prompt-free approach to the attack on SAM. I am curious about the types of prompts that may be more advantageous in crafting effective UAPs during the attack generation process. Additionally, could the authors provide an explanation regarding the selection of prompts?
>
> **A4**: What a valuable question! In the manuscript, we provide an evaluation of the attack performance of UAPs created using point and box prompts on SAM across three segmentation modes (see Tabs 1, 2, and Fig. 5). The results in Tab. 1 and Fig. 5 show that the attack performance using point prompts does not differ significantly from that using box prompts. However, Tab. 2 indicates that UAPs created with box prompts exhibit better transferability, which may be due to the additional information they provide (Line 255 - Line 260).
>
> ------
>
> **Q5**: The paper relies on heuristic research and lacks a corresponding theoretical framework for analysis.
>
> **A5**: We acknowledge that the paper primarily relies on heuristic research and does not provide a complete theoretical framework. Our intent is to explore and validate new approaches for assessing the robustness of SAM and its variants through heuristic methods (see Sec. 3.2), which lays the groundwork for future theoretical modeling. We will include a discussion of these limitations in the "Conclusions, Limitations, and Broader Impact" section. In future work, we plan to further develop and refine the theoretical framework to systematically analyze our findings.

---

> > ### Comment · Reviewer_dgw7 · 2024-08-07
> > **Response from the reviewer**
> >
> > After carefully reviewing the comments from the other reviewers and the author's rebuttal, I find that all of my concerns have been adequately addressed. Therefore, I have decided to raise my score to 8.

---

> > > ### Author Response · Authors · 2024-08-08
> > > **Response to Reviewer  dgw7**
> > >
> > > Dear Reviewer dgw7,
> > >
> > > Thank you for your positive feedback! We would like to express our deep gratitude for your dedicated time and effort in reviewing our manuscript. If you have any further questions, please leave us new comments.
> > >
> > > Best regards,
> > >
> > > The Authors

---

### Official Review · Reviewer_YLRY · 2024-07-10

**Soundness:** 3
**Presentation:** 3
**Contribution:** 3
**Rating:** 6
**Confidence:** 5

**Summary:**

This paper introduces DarkSAM, a universal adversarial attack against the Segment Anything Model and its variants. DarkSAM aims to prevent these models from successfully segmenting objects within images. The experimental results demonstrate the effectiveness and transferability of the proposed method.


I have read the response of the authors and comments of other reviewers, I decide to keep my weak accept score.

**Strengths:**

1.This paper introduces a new universal adversarial attack framework for prompt-guided image segmentation models.
2.The combination of spatial and frequency domain attacks is a sophisticated approach that demonstrates a good understanding of SAM.
3.This paper is well-written. Following the introduction, I can easily understand the goal of this paper.

**Weaknesses:**

1.The related work can be improved. This paper could benefit from an expanded discussion on adversarial attacks targeted at traditional segmentation models.
2.Lack of specific explanation. This method is novel and interesting but I’m curious about the reason why it works. What is the exact process for determining random prompts, and does this method ensure coverage of all potential attack targets?
3.The experimental results lack error bars. Repeating the experiments a few times and reporting the results with error bars would make the findings more convincing.

**Questions:**

See Weaknesses

**Limitations:**

The authors adequately addressed the limitations and potential negative societal impact of their work.

---

> ### Author Rebuttal · Authors · 2024-08-05
>
> # Responses to Reviewer YLRY #
>
> ------
>
> **Q1**: The related work can be improved. This paper could benefit from an expanded discussion on adversarial attacks targeted at traditional segmentation models.
>
> **A1**: Thank you for the constructive feedback! We will include a more comprehensive discussion on adversarial attacks targeting traditional segmentation models in the revised version.
>
> ------
>
> **Q2**: Lack of specific explanation. This method is novel and interesting but I’m curious about the reason why it works. What is the exact process for determining random prompts, and does this method ensure coverage of all potential attack targets?
>
> **A2**:  We have outlined the process of the shadow target strategy in Fig. 2 of the manuscript. For each image, we randomly generate *k* prompts, obtain the outputs from SAM for each, and then merge these results into a "Blueprint" to be used as the attack target (Line 138). The goal of this strategy is not to ensure coverage of all potential attack targets, but rather to maximize the creation of shadow targets for generating UAPs. We are willing to refine the relevant statements in the revised version.
>
> ------
>
> **Q3**: The experimental results lack error bars. Repeating the experiments a few times and reporting the results with error bars would make the findings more convincing.
>
> **A3**: Thank you for the valuable suggestions. We test the attack performance of the proposed method on SAM using three different random seeds across four datasets. The experimental setup is consistent with that described in Sec. 4.2 of the manuscript. The results in Tab. R1 demonstrate that our method consistently exhibits robust attack performance.
>
> Table R1: The mIoU (%) of DarkSAM under different settings
> | Prompt | Surrogate | ADE          | COCO         | CITY         | SA-1B        |
> | ------ | ------- | ------------ | ------------ | ------------ | ------------ |
> | Point  | Clean   | 65.07 ± 0.25 | 63.04 ± 0.14 | 50.43 ± 0.24 | 77.18 ± 0.05 |
> |        | ADE     | 0.73 ± 0.33  | 2.91 ± 0.58  | 1.03 ± 0.64  | 4.50 ± 1.37  |
> |        | COCO    | 0.16 ± 0.13  | 0.41 ± 0.38  | 0.27 ± 0.25  | 0.93 ± 0.77  |
> |        | CITY    | 7.64 ± 1.95  | 0.13 ± 0.06  | 16.02 ± 3.57 | 11.68 ± 5.32 |
> |        | SA-1B   | 0.60 ± 0.41  | 0.01 ± 0.01  | 2.18 ± 1.66  | 0.08 ± 0.03  |
> | Box    | Clean   | 74.49 ± 0.24 | 79.10 ± 0.08 | 64.84 ± 0.41 | 89.24 ± 0.09 |
> |        | ADE     | 6.57 ± 0.90  | 13.27 ± 1.35 | 1.98 ± 0.21  | 19.19 ± 2.21 |
> |        | COCO    | 1.68 ± 0.47  | 3.21 ± 0.49  | 2.11 ± 0.82  | 9.44 ± 1.26  |
> |        | CITY    | 20.35 ± 5.76 | 29.15 ± 7.64 | 0.78 ± 0.53  | 23.32 ± 6.56 |
> |        | SA-1B   | 12.20 ± 3.20 | 22.40 ± 4.62 | 4.31 ± 2.45  | 5.56 ± 0.95  |
> |

---

### Decision · Program_Chairs · 2024-09-25

**Decision:**

Accept (poster)

**Comment:**

This work was accepted by all three reviewers (Reviewer YLRY, Reviewer dgw7, Reviewer pM4H). The authors’ rebuttal responses have addressed many of the reviewers’ concerns. I am inclined to accept this work based on technical contribution and strong experimental evaluation.